# Clinical, immune and genetic risk factors of malaria-associated acute kidney injury in Zambian children: A study protocol

Chisambo Mwaba[1,2]*, Sody Munsaka[3⚬], David Mwakazanga[4⚬], David Rutagwerae[5⚬], Owen Ngalamika[6⚬], Suzanna Mwanza[7⚬], Mignon McCulloch[8⚬], Evans Mpabalwani[1,2⚬]

1 Department of Paediatrics and Child Health, School of Medicine, University of Zambia, Lusaka, Zambia,
2 Department of Paediatrics, University Teaching Hospitals-Children's Hospital, Lusaka, Zambia,
3 Department of Biomedical Sciences, School of Health Sciences, University of Zambia, Lusaka, Zambia,
4 Public Health Department, Epidemiology and Statistics Unit, Tropical Diseases Research Centre, Ndola, Zambia, 5 Kaposi's Sarcoma Molecular Laboratory, Paediatric Centre of Excellence, University Teaching Hospitals-Children's Hospital, Lusaka, Zambia, 6 Dermatology and Venereology Division, Department of Internal Medicine, University Teaching Hospital, University of Zambia School of Medicine, Lusaka, Zambia, 7 Department of Paediatrics and Child Health, Chipata Central Hospital, Chipata, Zambia, 8 Division of Paediatric Nephrology, Red Cross War Memorial Children's Hospital, University of Cape Town, Cape Town, South Africa

⚬ These authors contributed equally to this work.
* chisambo.mwaba@unza.zm

**Data Availability Statement:** No datasets were generated or analysed during the current study. All

## Abstract

### Background

Acute kidney injury (AKI) affects nearly half of children with severe malaria and increases the risk of adverse outcomes such as death and poor cognitive function. The pathogenesis and predictors of malaria-associated acute kidney injury (MAKI) are not fully described. This study aims to determine the clinical, immune, and genetic correlates of risk to AKI in Zambian children admitted with malaria. In addition, we intend to assess a modified renal angina index (mRAI), kidney injury molecule-1 (KIM-1), neutrophil gelatinase-associated lipocalin (NGAL), and soluble urokinase receptor (suPAR), when done on the first day of admission, for the ability to predict AKI two days later (day 3) in children admitted with malaria.

### Methods

This is an unmatched case-control study with a nested prospective observational study. A case-to-control ratio of 1:1 is used and 380 children with malaria and aged less than 16 years are being recruited from two hospitals in Zambia. Eligible children are recruited after obtaining written informed consent. Recruitment occurs during the malaria season and began on 6th March 2024 and will continue until July 2025. AKI is defined using the 2012 KIDGO AKI creatinine criteria, and cases are defined as children admitted with malaria who develop AKI within 72 hours of admission, while controls are children admitted with malaria but with no AKI. Serum creatinine is collected on Day 1 within 24 hours of admission, on Day 3 and then again on discharge or day 7, whichever comes sooner. Baseline biomarker

relevant data from this study will be made available upon study completion.

**Funding:** CM received a PHD scholarship from the Zambia ministry of Technology and Science. The sponsor did not play any role in study design, data collection, decision to publish, or preparation of the manuscript.

**Competing interests:** No authors have competing interests.

concentrations will be determined using the Luminex multiplex Elisa system or high-sensitivity ELISA. SPSS version 29 will be used for data analysis. Descriptive statistics and inferential statistical tests will be run as appropriate. A $p \leq 0.05$ will be considered as significant. The sensitivity, specificity, and estimates of the area under the curve (AUC) for the renal angina score will be determined.

## 1.0 Introduction

### 1.1 Background

Worldwide, malaria is still responsible for nearly a quarter of a million under-five deaths annually, and although Zambia contributes a mere 1% to total worldwide cases, malaria is the leading cause of mortality in the country [1, 2]. Many of these deaths are attributable to the development of organ dysfunction, including acute kidney injury (AKI). AKI, which manifests in approximately 10–50% of children with malaria, increases the risk of mortality sixfold, and has long-term adverse effects such as chronic kidney disease (CKD) and cognitive impairment [3–7]. At Lusaka's University Teaching Hospitals-Children's Hospital (UTH-CH), malaria stands as the predominant cause (60%) of AKI in children undergoing dialysis, with nearly 20% of these cases progressing to CKD [8]. The development of CKD and end-stage kidney disease (ESKD) in these patients, is life-limiting due to constraints in access to kidney replacement therapy and transplantation [9]. Yet despite this state of affairs, the epidemiology, immune and genetic risk factors of MAKI are incompletely described in our population.

Risk factors in MAKI can be categorized as either biological or clinical. Biological risk factors are the various molecules involved in MAKI pathogenetic mechanisms such as cytoadherence, immune dysregulation, heme nephrotoxicity, and increased oxidative stress [5, 6, 10–12]. The relative importance of a given risk factor in a given population may be modulated by peculiarities in local medical practice, access to healthcare resources, social determinants of health, shifting patterns of malaria transmission intensity, and the effect of genetic variability on the functionality of the various immune molecules that are involved in MAKI pathogenesis.

Children with malaria are at increased risk of dehydration because they often present with excessive fluid losses resulting from diarrhea, vomiting, increased insensible water losses, and reduced ability to ingest fluids due to poor appetite, nausea, or decreased levels of consciousness that may accompany the illness. Conroy et al. found that a urea/creatinine ratio consistent with pre-renal AKI was prevalent in children with MAKI [13]. Other investigators have reported comparable findings, with the presence of shock being more likely in children with MAKI [14]. The effect of fluid losses on the kidney may be further worsened by limited access to an appropriate level of health care and cultural practices such as seeking out traditional treatments prior to presentation and use of nephrotoxins such as herbal medications and aminoglycosides [14, 15].

Another potential biological nephrotoxin is the oxidant, cell free haemoglobin [3, 12, 16, 17]. Elphinstone et al. showed that the haem/hemopexin ratio was higher in paediatric malaria patients with stage 3 AKI compared to those without [18]. Similarly, increased serum markers of hemolysis (bilirubin, lactate dehydrogenase) have been shown to correlate to greater odds of MAKI [14]. This is of particular concern in patient populations who may already have excessive intravascular haemolysis from underlying hemolytic genetic defects or indeed malaria patients with an increased parasite load (pfHRP2) who are at risk of greater haemolysis.

It is thought that vascular obstruction resulting from red cell sequestration in the renal circulation contributes to compromised microvascular perfusion, which in turn may contribute to MAKI pathogenesis [5, 12]. *Plasmodium falciparum* generates a cell-surface molecule known as PfEMP1, which facilitates the adhesion of infected red cells to endothelial cells in a process known as sequestration [19, 20]. Various studies have shown a link between heightened sequestration and parasite load to higher risk of MAKI. Indirect evidence includes a correlation between markers of heightened sequestration such as *Plasmodium falciparum* histidine-rich protein 2 (*Pf*HRP2) and MAKI, while more direct evidence has been drawn from human autopsy and experimental studies [14, 21–23]. Furthermore, cytoadherence causes endothelial activation, which in turn causes cytoadherence to worsen. Thus, a vicious cycle is established in which endothelial activation results from cytoadherence and vice versa [10, 12, 24]. Some studies have found that increased markers of endothelial activation, increased circulating soluble cell adhesion molecules, as well as increased monocyte activation are linked to a greater risk of organ dysfunction, including MAKI [25–29].

Furthermore, polymorphisms in the genes coding for these various immune molecules may influence gene function with resultant alteration in amounts or function of the gene products [30, 31]. Numerous studies have suggested a link between immune molecule gene polymorphisms and various severe malaria syndromes [32–35]. However, most of these studies lacked statistical power to detect associations with MAKI due to heterogeneous samples encompassing various forms of severe malaria.

There is limited data evaluating the utility of these various immune molecules, or indeed the numerous emerging AKI biomarkers such as kidney injury molecule-1 (KIM-1), neutrophil gelatinase-associated lipocalin (NGAL), and soluble urokinase receptor (suPAR), in the prediction of MAKI [36, 37]. Additionally, although some clinical prediction rules, such as the Renal Angina Index (RAI), have been formulated, these have mostly been tested in general paediatric intensive care unit (PICU) populations and have not been validated in children with malaria [38]. RAI utilizes risk factors for AKI and early clinical signs of kidney injury to create a composite score with a score greater than 8 out of 40 being predictive of increased AKI risk [38]. Some studies have attempted to modify the RAI for use in different patient populations [39–41].

We hypothesize that various clinical factors, serum concentration of immunological molecules, as well as genetic polymorphisms of selected immune factors, are correlated to the risk of developing AKI in children admitted with malaria and can therefore be used for risk prediction. Specifically, we hypothesize that 1) delayed access to care, hyperparasitaemia, increased oxidative stress, presence of exaggerated haemolysis and dehydration increase the risk of MAKI, 2) MAKI is associated with higher concentrations cytokines or serum markers of endothelial activation, 3) the Renal Angina Index and/or its modification, NGAL, suPAR, and KIM-1, when used in severe malaria, may be useful in predicting severe AKI 48 hours later (Day 3) and 4) ten (10) immune molecule single nucleotide gene polymorphisms (SNPs) are associated with an altered risk of AKI in children admitted for malaria.

The purpose of this study is to 1) to determine the correlation between clinical risk factors (hyperparasitaemia, delayed access to care, urea/creatinine ratio> 20, increased oxidative stress, increased serum bilirubin and low haemoglobin) and MAKI, 2) determine the correlation between plasma pro- and anti-inflammatory cytokine concentrations and MAKI, 3) to assess the relationship between ten (10) immune molecule gene polymorphisms namely interleukin (IL)-10 (rs1800871, rs1800871), tumor necrosis factor-alpha (TNF-α) (rs361525, rs1800629), Type I Interferon alpha receptor 1 (IFNAR1) (rs2843710, rs2243594), haem-oxygenase-1(HMOX1) (rs7285877), nitric oxide synthetase 2(NOS2), Transforming growth factor beta-2 (TGF-Beta 2) (rs4846478) and toll-like receptor- 4(TLR4) (rs4986791) and the MAKI

phenotype, 4) to determine the utility of the Renal Angina Score in predicting MAKI and access whether its modification with population specific predictors will lead to enhanced performance and finally, 5) to determine the predictive value of novel biomarkers (NGAL, suPAR, KIM-1) in MAKI. We hope that the proposed work will provide evidence that will improve early diagnosis of MAKI and consequently result in the adoption of protocols with the institution of remedial treatments and referral of children in a timelier manner.

## 2.0 Materials and methods

### 2.1 Study design

The project is divided into two sub-studies, namely 1) Sub-study 1 which answers objectives 1, 2 and 3, and is an unmatched case-control study with a case:control ratio of 1:1. An unmatched design was chosen because this is an exploratory study. However, during analysis of data for the immune molecules, patients will be analysed based on age groups. Sub-study 2 answers objectives 4 and 5 and is a nested prospective observational study that recruits participants from the control arm of sub-study 1 (Fig 1).

This is a case-control study with a nested prospective observational study. Participant serum creatinine will be determined on Day 1 and Day 3. Patients meeting the KDIGO AKI

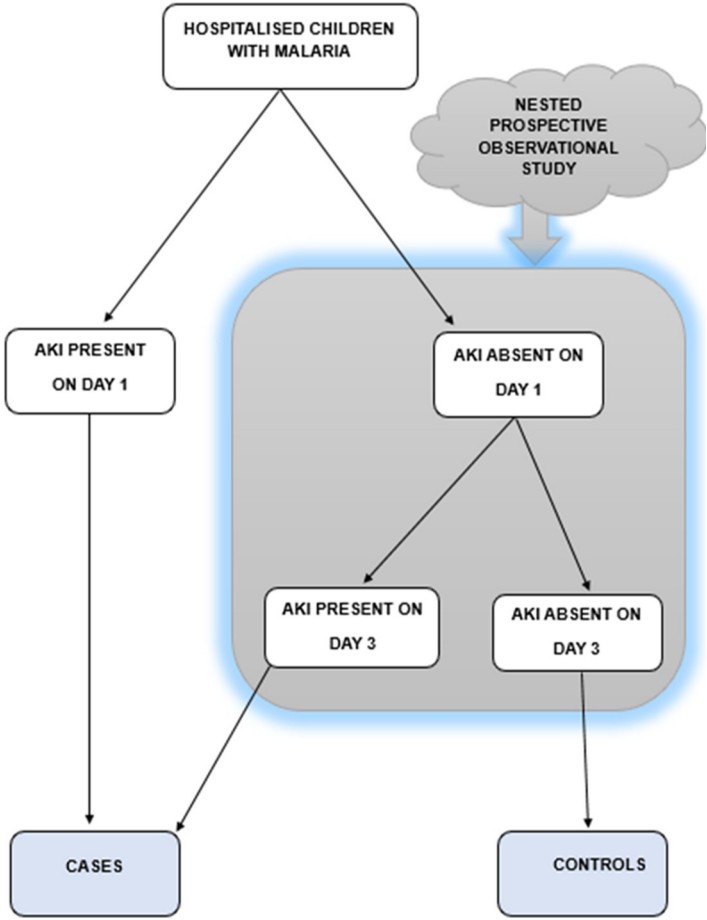

**Fig 1. Study design and recruitment plan.**

creatinine criteria will be classified as cases, while those who do not meet the KDIGO creatinine AKI criteria will be classified as controls. The nested prospective observational study draws its study population from the cohort of malaria patients with absent AKI on day l (Fig 1). Serum creatinine will be repeated on Day 3 to determine if KDIGO stage 2/3 (severe) AKI will have developed on Day 3.

## 2.2 Study setting

The study participants are drawn from Chipata Central Hospital (CCH) in Chipata town of the Eastern province of Zambia and from the University Teaching Hospitals Children's Hospital (UTHs-CH) in the capital city of Zambia, Lusaka.

Chipata town is located about 600 km from Lusaka and is the provincial capital of the Eastern province, which is a medium malaria transmission zone (parasite prevalence among children in the Eastern Province is 21%) [43]. CCH is a tertiary hospital with the four major specialties of paediatrics, internal medicine, surgery, and obstetrics and gynecology available. The hospital has a capacity of 500 beds and is an internship- and fellowship-training site for all the above-mentioned specialties. The paediatrics department has a capacity of 70 beds with access to a level II intensive care unit (ICU) that admits neonates, children, and adults with both medical and surgical conditions. The selection of this site was based on a prior study conducted at UTH, revealing that CCH ranked as the second most frequent hospital referring children with MAKI for treatment at UTHs-CH, nephrology unit. Additionally, the chosen site provides robust support for the study. The site has good laboratory support, including the presence of an ultra-low freezer (-80°C) and the capacity to conduct PCR.

The UTH-CH is the leading tertiary paediatric hospital in Zambia, located in Lusaka district, which is a very low malaria transmission zone [42]. The hospital receives referrals from across the country and has the older and larger of the two pediatric dialysis centers in the country. The hospital has a capacity of 350 beds. The nephrology unit evaluates about 300 children per year, and a third of these have AKI. The site was chosen to capture the more severe forms of MAKI that tend to be referred from across the country. The site has access to molecular laboratories including, the Kaposi's Sarcoma (KS) molecular laboratory which has subzero freezers, PCR machines, sequencers, flow cytometers, facilities for gel electrophoresis, and staff trained in these techniques.

## 2.3 Study population

Children aged less than 16 years who are admitted for malaria (modified CDC 2014 definition) at the UTHs-CH and CCH constitute the target population. Recruitment began on 6th March 2024 and will continue until July 2025. For sub-study 2, the nested prospective observational study, the study population is drawn from the cohort of malaria patients with no AKI on day l (Fig 1).

## 2.4 Inclusion and exclusion criteria

To be included in the study, a child must be between 6 months and 16 years old, have parasitological evidence of malaria, and be admitted to either UTHs-CH or CCH. The upper age limit for admission to the pediatric services at the two hospitals is 16 years.

Children who are known to have CKD and children who have received dialysis prior to admission for the episode of AKI are excluded. Dialyzed patients are excluded because serum levels of various biomarkers may be distorted during dialysis.

## 2.5 Study definitions

Any child with a history of fever exhibiting signs or symptoms consistent with malaria (fever, chills and sweats, headaches, muscle pains, nausea, vomiting, fatigue, anemia, coma, seizures, focal neurological signs, respiratory distress) who shows evidence of plasmodium infection by a) detection of malaria by rapid diagnostic antigen (RDT) testing without confirmation by microscopy or nucleic acid testing e.g., in this study [43]. OR b) detection of parasites by microscopy on blood films will be defined as having malaria.

A child aged 6 months to 16 years old who meets the KIDGO 2012 AKI criteria and who has malaria as per the modified CDC malaria case definition is categorized as having MAKI. A case is a child 6 months old to 16 years of age with MAKI detected within the first three days of admission while a control is defined as a child aged 6 months to 16 years of age with malaria but with NO MAKI within the first three days of admission.

## 2.6 Sample size estimation and sampling methods

The study plan is to recruit a total of 380 patients, with 190 being cases and 190 being controls. A rate of 10% for missing data is factored in. This sample size will provide 80% power, with a statistical significance of 5%, to detect effect sizes equivalent to those reported in previous studies [13, 14, 44–52]. Supplementary information (S1 File) provides details for both sub-study 1 and sub-study 2. This is an exploratory study therefore, no matching of cases to controls will be conducted at recruitment. However, during laboratory analysis of the various immune molecules, 100 cases will be matched to 100 controls based on age groups (less than 5 years, 5–10 years, and greater than 10 years), with 20 children assigned to each age group for both cases and controls, to account for the well-known effect of age on immune molecule serum concentrations. Consecutive sampling is being used.

In the following sections the recruitment and laboratory procedures, the study variables, and the data analysis plans for sub-study 1 (section 2.6) and the nested prospective sub-study 2 (section 2.7) have been outlined.

## 2.7 Sub-study 1

**2.7.1 Sub-study 1 recruitment procedure.** Screening of children with malaria is conducted by the study nurse. The purpose of the study is explained to the legal guardians or parents of all potential participants. Written informed consent and assent (children > 8 years if fully conscious) where appropriate, is obtained after explaining the purpose of the study to potential participants. The information sheet, consent, and assent forms are available in the predominant languages of these regions, namely English, Bemba, Nyanja, and Tonga, and are administered according to the preferred language of communication chosen by potential participants. Each participant is then assigned a unique study number. A pre-designed data collection form on Redcap TM is used to collect participant demographic and clinical data.

Children are recruited from both study sites. The recruitment procedure is illustrated in Fig 1. The case definition of MAKI is the development of AKI (KDIGO 2012 creatinine criteria) within the first three days of admission. The patient's creatinine is collected on recruitment (Day 1, within the first 24 hours of admission) as well as on day 3 of admission. Malaria patients meeting the definition of MAKI based on creatinine results collected at recruitment on day 1 or day 3 are assigned to the case arm of the study. Malaria patients who do not meet the definition of MAKI based on serum creatinine results from samples collected on day 1 and day 3 are assigned to the control arm of the study. Additionally, all children referred to the study site with a diagnosis of MAKI already made are also recruited as cases after confirmation of blood results at the study site.

Specimens for serum urea, bilirubin, potassium, sodium, haemoglobin, and platelet count are also collected. Finally, also being collected at baseline are specimens for kidney biomarkers, cytokines, adhesion molecules, pfHRP2 and malondialdehyde (MDA).

A venous blood aliquot, which is computed based on the child's age and weight, is drawn, ensuring that the amount for any single blood draw does not exceed the maximum allowable volume, set at 2.5% of the total blood volume, following widely used guidelines for both clinical and research purposes [53]. The collected whole blood is placed in a heparinized vacutainer, 2 EDTA vacutainer, and in a plain tube. The specimen in the plain tube vacutainer will be left at room temperature for 10 minutes to facilitate clot formation. Subsequently, this sample undergoes centrifugation at 3000 rpm for at for 5 minutes, after which the supernatant is stored at -80˚C. The sample in the heparinized container similarly undergoes centrifugation at 3000 rpm for 5 minutes. The supernatant is also stored in a cryovial and stored at -80˚C for onward transport and processing at UTH-CH.

**2.7.2 Sub-study 1 laboratory procedure.**   All the specimens kept at -80˚C are transported to the UTH-CH on dry ice for analysis. The Beckman Coulter AU 480 analyser will be employed to assess blood parameters, including creatinine, sodium, potassium, bilirubin, and urea, as per manufacturer instructions. Blood platelet count and haemoglobin are determined at each respective study site using the Sysmex XT 4000i haematology analyser.

To conduct the cytokine analysis the frozen serum will be thawed. A bead-based multiplex assay (cytometric bead array) will be done on a Luminex™ platform, at the KS molecular laboratory at UTH-CH, and will be used to analyze the serum cytokine concentrations. The manufacturer's protocol will be followed to run the assay.

A calibration curve for calculating the concentration of cytokines will be created using the manufacturer's provided lyophilized standards in the kits, and the cytokine concentrations in samples will then be deduced from standard curves. Because immunoassays generate a sigmoid-shaped curve, logistic regression mathematical modelling is used to allow curve fitting beyond the linear range of the curve. This facilitates subsequent determination of the Y-intercept and gradient. The lower limit of detection for the various cytokines will be determined. If readings are above the upper limit of detection, dilutions of 1:4, 1:8, 1:16, and so on will be done until a negative result, then the dilution factor will be used to determine the actual concentration [49].

For quantification of oxidative stress, the MDA NWLSSTM malondialdehyde assay will be used. All the laboratory analytes in this study are shown in Table 1.

**Table 1. Study analytes.**

| Biomarker | Specimen | Test |
|---|---|---|
| **Interleukin 18**<br>**Interleukin 1**<br>**Interleukin 10**<br>**Interleukin 6**<br>**Interleukin 12**<br>**Soluble TNF-α**<br>**IFN-γ** | Plasma | Cytometric bead array |
| **Angiopoietin 1**<br>**Angiopoietin 2**<br>**Endothelial cell protein C receptor (EPCR).**<br>**Intercellular adhesion molecule 1 (ICAM-1)**<br>**Vascular cell adhesion protein-1 (VCAM)** | Plasma | Cytometric bead array |
| **Malondialdehyde (MDA)** | Plasma | MDA NWLSSTM malondialdehyde assay (commercial oxidative stress kits) |
| ***Plasmodium falciparum* histidine-rich protein 2 (pfHRP2)** | Blood | ELISA |

ELISA = enzyme-linked immunosorbent assay

Twelve polymorphic markers are proposed for investigation in this study. They have been chosen based on previous associations in the literature to other forms of severe malaria or other AKI phenotypes with a postulated similarity in pathophysiology to MAKI.

Whole blood is collected onto special filter paper cards as dried blood spots for later genomic DNA extraction [54]. Genomic DNA will be extracted using a commercial DNA purification kit (QIAwave DNA Blood and Tissue Kit by Qiagen TM). Polymerase Chain Reaction (PCR) will be used for DNA amplification. We will use primers as described in previous experiments, and for those SNPs where we could not find references in the literature, we searched NCBI gene to obtain information on the location of the target, then we obtained the sequence of the area of interest in FASTA format and used the NCBI primer tool to design appropriate primers. The Qiagen™ Multiplex PCR kit (cat no. 206143) and primers (S2 File), and restricting enzymes for the 12 gene polymorphisms will be placed in a thermal cycler as per manufacturer protocol. Gel electrophoresis will then be conducted to determine the presence of the SNPs of interest, then for positive samples, the PCR products will be purified using a commercial kit, before subjecting them to Sanger sequencing (ABI 3500) at the School of Veterinary Medicine genomics laboratory, University of Zambia.

**2.7.3 Sub-study 1 variables.** The study collects information on the occurrence of MAKI in admitted malaria patients as the primary outcome variable. Serum creatinine is collected at discharge or if the patient is still hospitalized on day 7 to establish the renal outcome. Renal outcome is noted as a secondary outcome. The outcome is classified as normal renal function if eGFR is >120ml/1.73m2/min or abnormal for eGFR < 120ml/1.73m2/min. The Schwartz formula is used to estimate GFR [55].

The three categories of independent variables for this proposed study are socio-demographic, clinical, and laboratory variables. Socio-demographic factors encompass the patient's age, gender, place of residence (village/suburb), distance from the referring facility, the referring health facility, province of origin, and self-identified ethnic group. Ethnic group data is being collected for analyzing genetic information among participants.

Clinical independent variables include patient height/length, weight, hydration status (classified according to WHO as none, some, and severe) [56], vital signs (blood pressure, oxygen saturation, presence of edema), and Glasgow Coma Score upon admission. Additional clinical variables being collected include the reason for referral, identification of AKI at the referring facility, date of AKI identification if applicable, history of anuria/oliguria, and pre-admission seizure history. Other clinical variables comprise the duration of illness before admission, pre-admission medications and dosages, family history of renal disease, the patient's past medical history, and the number of blood transfusions in the current illness. Additionally, birth weight, length of hospital stay, admission to and duration of admission to PICU, and mortality, are documented.

Laboratory independent variables are the admission creatinine, admission estimated glomerular filtration rate (GFR), hemoglobin electrophoresis, admission hemoglobin, bilirubin, platelet count, and random blood sugar (RBS). Routine HIV testing is conducted for all pediatric patients upon admission, following an opt-out system for caregivers. Recorded HIV test results in the patient file are documented. Further laboratory results on admission include PfHRP2 levels, cytokine levels (IL-18, IL-1, IL-10, IL-12, soluble TNF-α, IFN-γ), adhesion molecule levels (Ang-1, Ang-2, EPCR, ICAM-1, VCAM), and markers of oxidative stress (MDA).

**2.7.4 Sub-study 1 data analysis.** Patient data will be analyzed utilizing IBM SPSS Statistics version 29 (IBM Corp., Armonk, NY, USA). Categorical variables will be presented as frequencies and percentages. For continuous variables, those with asymmetric distributions will be reported as medians and interquartile ranges (IQRs), while those with normal distributions

will be presented as means and standard deviations (SDs). The normality of continuous data will be assessed using the Shapiro-Wilk test.

To assess the strength of associations between various categorical exposure variables and the outcome (AKI), the chi-square test will be employed. In cases where any cell in the contingency table contains fewer than five expected observations, the Fisher's Exact test will be used.

For the association of normally distributed continuous variables, the independent t-test will be applied to compare means between the two groups. Non-normally distributed continuous data will undergo non-parametric testing, such as the Mann-Whitney U test or the Kruskal-Wallis test. Multivariate logistic regression will be conducted to address potential confounding factors. All statistical tests will be two-tailed, and a significance level of $p \leq 0.05$ will be considered statistically significant.

Analysis of immune molecule data will be conducted using GraphPad Prism software (Version 9, San Diego, CA, USA). Cytokine data will be represented as geometric means with a 95% confidence interval (CI) to illustrate the central tendency of cytokine levels. Significant differences between cases and controls will be assessed using either the t-test or the Mann-Whitney U test.

To evaluate the discriminatory power of each immune molecule between cases and controls, a Receiver Operator Curve (ROC) will be constructed. Multiple logistic regression analyses for each cytokine/adhesion molecule will be carried out to compare cases versus controls. The goodness of fit for the model will be assessed using the Hosmer-Lemeshow test.

The level of association of the various polymorphisms with MAKI will be determined. The clinical and demographic composition of the cases and controls will be compared using the Chi-square test or Fischer's Exact test. The level of correlation of the two groups to the presence of the SNPs of interest will be done using Chi-square and Fisher's exact tests. All loci will be assessed for the presence of Hardy-Weinberg equilibrium. A P-value of <0.05 will be considered statistically significant.

## 2.8 Sub-study 2 (Nested prospective cohort study)

**2.8.1 Sub-study 2 recruitment procedure.** This is a nested study utilizing a prospective observational study design. Recruitment is conducted as shown in Fig 1. Written informed consent and assent, where appropriate, are obtained after explaining the purpose of the study to potential participants as described in sub-study 1. Baseline scoring of the renal angina index (Table 2) within the first 24 hours after admission using a pre-designed data collection form is done. Fluid overload is determined as the difference in body weight at recruitment and at hospital admission divided by the weight at admission to the hospital expressed as a percentage [57] The patient baseline creatinine is collected at admission as described under the methods

**Table 2. The renal angina score.**

| RAI = Risk X Injury (score ranges from 1–40) | | | | | |
|---|---|---|---|---|---|
| **INJURY** | | | **RISK FACTOR** | | |
| % Drop in eGFR | % FO | **Score** | | Risk factor | **Score** |
| No change | $\geq 5\%$ | 1 | | PICU admission | 1 |
| 0–25% | $\geq 5\%$ | 2 | **X** | Stem cell transplantation | 3 |
| 25–50% | $\geq 10\%$ | 4 | | Ventilation or use of vasoactive agent | 5 |
| $\geq 0\%$ | $\geq 15\%$ | 8 | | | |

eGFR = estimated glomerular filtration rate, PICU = paediatric intensive care unit, RAI = Renal angina index

section for sub-study 1. The KDIGO 2012 creatinine criterion is used to define AKI, and severe AKI is defined as KIDGO 2012 stage 2 and 3 AKI as per previous studies [39].

For biomarkers (KIM-1, suPAR, NGAL), 4ml of whole blood is collected (same sample as for sub-study 1) in plain tubes and left at room temperature for 10 minutes until clot formation. Later, the sample is centrifuged. The supernatant is then stored at -80˚C as described in sub-study 1. Concentrations of AKI biomarkers (KIM-1, NGAL, suPAR) will be determined using a Luminex multiplex bead-based array or high-sensitivity ELISAs as outlined for sub-study 1.

**2.8.2 Sub-study 2 variables.** Sub-study 2's primary outcome is the occurrence of KDIGO stage 2 or stage 3 (severe) AKI on Day 3. Secondary outcome variables are death, whether a patient required dialysis or PICU admission, and the length of hospital stay.

The sub-study 2 independent variables are the composite Day 1 RAI score, the various patient baseline clinical features, and the baseline (day 1) serum concentrations of NGAL, KIM-1, and SuPAR.

**2.8.3 Sub-study 2 data analysis.** Data will be exported from RedCap™ to an Excel spreadsheet and then imported into an IBM™-SPSS™ version 29 (IBM Corp., Armonk, NY, USA) database for data analysis in SPSS version 29. Categorical data will be expressed as frequencies and percentages, while continuous data will be expressed as median and interquartile range if of asymmetric distribution while data that has a normal distribution will be expressed as means with standard deviation (SD). The Shapiro-Wilk test will be used to determine data distribution.

To test the association between continuous data and the outcome variable (severe AKI at Day 2 Yes/No), the independent t-test (parametric data) or Mann-Whitney test (non-parametric data) will be used. Categorical data will be tested for association to the outcome variable (severe AKI Yes/No), using the Chi-square or Fisher's exact test for cells containing less than five. A $p \leq 0.05$ will be considered as being statistically significant.

All variables showing an association with the bivariate form of the outcome variable (severe AKI 48hours later Yes/No) with a probability of less than 25% ($p < 0.25$), and all independent variables previously shown to have an association with the outcome variable (severe AKI on day 3 of-admissions Yes/No) from the literature will be used to construct a multivariate logistic regression model.

The quality of the multivariable models will be assessed using the likelihood-ratio (LR) chi-square test on the models for the global null hypothesis that none of the added potential predictors predict the dependent variable, the Hosmer and Lemeshow (H&L) goodness-of-fit test, and the C-statistic.

The utility of the renal angina score for predicting MAKI will be assessed using sensitivity, specificity, negative and positive predictive values (NPV and PPV) and estimates of the area under the curve (AUC). Local risk factors as identified from the cohort using multivariate logistic regression will be incorporated into the RAI to test if this will improve performance. Youden's index will be used to determine the cut-off point for the modified RAI, and curve fitting techniques will be utilized as described in previous studies by Matsuura et al. [39–41] The AUC of the modified RAI and the AUC obtained from the original RAI will be compared using the z-test (non-parametric).

## 2.9 Ethical considerations

This research is being conducted in compliance with guidelines for good clinical practice and ethical standards as espoused in the Declaration of Helsinki and the International Declaration on Human Genetic Data.

**2.9.1 Permissions.** Ethical clearance was sought and obtained from UNZABREC (REF. No. 4269–2023) and NHRA. Further, the hospital management at CCH and UTHs-CH have granted permission to conduct the study in the two sites, respectively.

**2.9.2 Informed consent.** The study nurse explains the purpose of the study to parents/guardians of prospective study participants in the language of the participants' choice, utilizing an interpreter where necessary (the interpreter will also co-sign the consent form) and using the study information sheet for guidance. The study nurse further explains that participation is voluntary and that participants would be free to withdraw at any time and without explanation if they so wish and that they will not suffer any punitive measures as a result. The parents/guardians are offered a copy of the study information sheet in a language of their choice to read through and follow along. They are also given an opportunity to seek further clarifications or ask questions, and then the study nurse invites them to participate in the study.

If the parent guardian agrees to participate, they are asked to sign the written consent form, which is available in Bemba, English, Lozi, Nyanja, and Tonga. The signature is witnessed by a person of the parents' choice. The study nurse takes a picture of the signed consent form and uploads it to the study database, gives a copy to the parent/guardian, and keeps a copy in a file kept at the study site in the research office at the site.

**2.9.3 Assent.** For children whose parent/guardian has given written informed consent, and the child is older than 7 years and able to understand the information presented and is medically able to assent, they are asked to assent by the study nurse in the presence of their parent/guardian.

**2.9.4 Confidentiality.** The interview is conducted in a corner or room away from the main ward. Each participant is assigned a unique study number that is used with the stored data and during the laboratory processing of samples. The screen book with participant data, which includes names, is kept in a locked cabinet in the site research office. Once the study concludes the logbooks will be held by the overall PI at UTH-CH in a locked cabinet at UTH-CH.

**2.9.5 Description of risks.** There are no additional risks associated with participation in the study. Participants experience mild pain during phlebotomy. The risk of infection to the injection site during phlebotomy is reduced by sterilizing the site of the injection as per standard medical/hospital procedure. These procedures are all conducted by trained health workers (nurses or doctors).

**2.9.6 Emergency care and insurance for research-related injuries.** This is an observational study, so no interventions will be made on participants, and all procedures, such as phlebotomy, do not represent added risk beyond that which participants undergo during the process of receiving the standard care offered by the hospital. Thus, no insurance is offered to participants.

**2.9.7 Description of benefits and anticipated gain in scientific knowledge.** There are few direct benefits to participants. At times hospitals do not have functional laboratory consumables to conduct regular blood tests, but during the study for all participants, baseline blood results are done and made available to the attending doctor. A transport refund is also offered if the collection of discharge data may delay the departure of the patient.

Much of the anticipated benefit is long-term. Information gleaned from this work will shed light on the risks of MAKI in the local Zambian environment and may help clinicians and policymakers formulate strategies to prevent MAKI in the future.

Also, if the various biomarkers evaluated prove to be able to accurately predict MAKI before it is clinically evident, they may be used to design point-of-care tests that will make diagnosis of MAKI faster and earlier and allow implementation of treatments to mitigate the effects of MAKI in children.

## 3.0 Discussion

### 3.1 Study limitations

In areas of high malaria endemicity, a positive slide, RDT or even the presence of a positive nucleic acid test does not definitively prove that a patient has clinical malaria, as many children may have asymptomatic infection along with other conditions that could explain the clinical presentation [58, 59]. This may lead to an over-estimation of MAKI cases.

Secondly, since the proposed study is not prospective from the point of infection, the recruited patients are at different stages along the disease pathophysiology. This may influence the levels of the various biomarkers, which in turn may act as a confounder, affecting the significance that may be attached to the determined risk factors.

Moreover, the significance of specific SNPs in relation to the disease varies frequently across diverse populations and is notably influenced by ethnicity. Many of the SNPs employed in this study were originally characterized in diverse populations, which may result in a lack of association with MAKI. This might lead to a false conclusion that polymorphisms associated with these immune factors have no effect on the severity of renal involvement in malaria.

It is an accepted fact that the relationship between genetic variation and expression of complex syndromes such as AKI is complex because most of the gene products work within networks in which redundancy, compensatory mechanisms, and feedback mechanisms come into play. Thus, the lack of association between an SNP and AKI in an in vivo setting may just reflect this fact.

### 3.2 Data management and quality assurance

Data is captured using a digital form in RedCapTM. Data will be exported to a Microsoft Excel version 16.0 (Microsoft Corp., Redmond, WA, USA) spreadsheet and then imported into an IBM TM-SPSS TM version 25 (IBM Corp., Armonk, NY, USA) database for storage and initial data analysis. The database will be securely stored on a password-protected computer, with access restricted to essential research team members, such as the statistician.

Prior to the commencement of participant recruitment, all research team members underwent training on data collection standards, conducted by the Principal Investigator (PI). Throughout the study, the PI will perform weekly reviews of all data entry forms to ensure completeness and accuracy. Any identified errors or missing data will be addressed promptly by consulting the responsible research team member for correction. All team members have been briefed on the proper protocols for specimen collection, transportation, and handling.

Specimens designated for molecular and genetic analysis will be transported on dry ice in a cooler box from the Clinical center to the KS laboratory, where they will be stored for long-term analysis. The laboratories involved in specimen processing include the KS molecular laboratory and the University of Zambia Veterinary School genetics laboratories. Both laboratories are accredited and recognized as national reference laboratories.

Procedures for specimen handling and processing will adhere strictly to established standards. These standards include appropriate labeling, documentation, and storage conditions to maintain specimen integrity throughout the study.

### 3.3 Dissemination plan

The results of this study will be used to write a thesis in partial fulfilment for the award of the degree of Doctor of Philosophy (PhD) at the University of Zambia. Reports will be presented to the medical library and the university, and a soft copy will be available online. Findings will be published in peer-reviewed journals.

**Table 3. Gannt chart.**

| Activity | 2023 | | | | 2024 | | | | 2025 | | | | 2026 | | | |
|---|---|---|---|---|---|---|---|---|---|---|---|---|---|---|---|---|
| | Q1 | Q2 | Q3 | Q4 | Q1 | Q2 | Q3 | Q4 | Q1 | Q2 | Q3 | Q4 | Q1 | Q2 | Q3 | Q4 |
| **Research proposal Development** | ▒ | ▒ | | | | | | | | | | | | | | |
| **Ethics and NHRA approvals** | | | ▓ | ▓ | | | | | | | | | | | | |
| **Sample Collection** | | | | | ▒ | ▒ | ▒ | ▒ | ▒ | ▒ | ▒ | | | | | |
| **Laboratory Analysis** | | | | | | | | | | | | ▓ | ▓ | ▓ | ▓ | |
| **Review paper Submitted** | | | | | ▒ | ▒ | | | | | | | | | | |
| **Paper 1 Submitted** | | | | | | | | | | ▓ | ▓ | | | | | |
| **Paper 2 Submitted** | | | | | | | | | | | | | ▒ | ▒ | | |
| **Paper 3 Submitted** | | | | | | | | | | | | | | ▓ | ▓ | |
| **Project wind down** | | | | | | | | | | | | | | | ▒ | ▒ |

Q = Quarter, NHRA = National Health Research Authority

## 3.4 Discarding of study materials

Leftover blood samples will be kept in a -80 C freezer in the KS molecular laboratory at UTH-CH for five years and may be used in future studies examining complicated malaria in children if permission is granted at that time from the ethical review board for an extension of the study.

## 3.5 Projected timeline

This study is projected to run from 2023 to 2026 as shown in Table 3.

# Supporting information

**S1 File. Effect sizes and estimation of sample size.**
(DOCX)

**S2 File. List of primer sequences to be used in the determination of the single nucleotide polymorphisms.**
(DOCX)

# Acknowledgments

We are grateful to Monica Mweetwa for reviewing the draft manuscript.

# Author Contributions

**Conceptualization:** Chisambo Mwaba, Sody Munsaka, Evans Mpabalwani.

**Funding acquisition:** Chisambo Mwaba.

**Methodology:** Chisambo Mwaba, Sody Munsaka, David Mwakazanga, David Rutagwerae, Owen Ngalamika, Suzanna Mwanza, Evans Mpabalwani.

**Project administration:** Chisambo Mwaba.

**Supervision:** Sody Munsaka, Mignon McCulloch, Evans Mpabalwani.

**Writing – original draft:** Chisambo Mwaba.

**Writing – review & editing:** Chisambo Mwaba, Sody Munsaka, David Mwakazanga, David Rutagwerae, Owen Ngalamika, Suzanna Mwanza, Mignon McCulloch, Evans Mpabalwani.

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
