## [Decision Letter · Decision Letter 0]

21 May 2024

PONE-D-24-13075Risk factors and predictors of malaria-associated acute kidney injury in Zambian Children: A study protocolPLOS ONE

Dear Dr. Mwaba,

Thank you for submitting your manuscript to PLOS ONE. After careful consideration, we feel that it has merit but does not fully meet PLOS ONE’s publication criteria as it currently stands. Therefore, we invite you to submit a revised version of the manuscript that addresses the points raised during the review process.

 This is an important study that will provide important insights into acute kidney injury in severe malaria. There is a clear need for high quality data from a variety of settings in Africa. Overall, the reviewers shared my enthusiasm and perspective regarding the importance of the study. However, they also raised several concerns regarding a lack of clarity on study definitions, organization and redundancy in the content, and rationale for the designed. I strongly encourage the authors to consider the reviewer comments and respond carefully and in detail.  At present, the manuscript is organized as a proposal to an IRB or higher education committee and does not align with standard approaches to write up study protocols. For example, rather than presenting the null hypotheses, objectives, and sample size calculations as bullet points and separate sections, I encourage the authors to summarize the information and present it instead as hypotheses, and objectives, and present the sample size calculation that was used to decide the final sample size. 

 Major revisions in the structuring of the content will be required for further consideration with the additional details requested by the reviewers. Please review the criteria on the website for guidance on how to structure protocol submissions and review published protocols for examples on what information to include. These guidelines must be followed for further consideration. https://journals.plos.org/plosone/s/submission-guidelines#loc-study-protocols While an observational study does not have specific checklists for submission like clinical trials or systematic reviews, I strongly encourage the authors to review the SPIRIT and PRISMA checklists for study protocols and the STROBE guidelines on presenting results for observations studies to understand requirements for presentation of information.  https://www.equator-network.org/  I wish you all the best with the requested revisions.

We look forward to receiving your revised manuscript.

Kind regards,

Andrea L. Conroy, PhD

Academic Editor

PLOS ONE

Reviewers' comments:

Reviewer's Responses to Questions

**Comments to the Author**

1. Does the manuscript provide a valid rationale for the proposed study, with clearly identified and justified research questions?

Reviewer #1: Partly

Reviewer #2: Yes

2. Is the protocol technically sound and planned in a manner that will lead to a meaningful outcome and allow testing the stated hypotheses?

Reviewer #1: Partly

Reviewer #2: Partly

3. Is the methodology feasible and described in sufficient detail to allow the work to be replicable?

Reviewer #1: No

Reviewer #2: Yes

4. Have the authors described where all data underlying the findings will be made available when the study is complete?

Reviewer #1: Yes

Reviewer #2: Yes

5. Is the manuscript presented in an intelligible fashion and written in standard English?

Reiewer #1: No

Reviewer #2: No

6. Review Comments to the Author

You may also provide optional suggestions and comments to authors that they might find helpful in planning their study.

Reviewer #1: Risk factors and predictors of malaria-associated acute kidney injury in Zambian

Children: A study protocol

The authors describe the methods for a prospective observational study in which they will match malaria associated AKI patients to those without malaria associated AKI, occurring by 72 hours form admission. will occur in a 1:1 scheme. There are no specific criteria on which the AKI cases will be matched with the non-AKI cases, other than the absence of AKI. The intent of the study is to characterize AKI risk factors, and use biomarkers for early AKI prediction with a modified renal angina index.

Overall, as written this protocol is very confusing. It could be better organized for improved flow and removal of redundancy. Specific comments below.

Title:

- To have “risk factors” and “predictors” in the title seems redundant, would select one of them.

Abstract:

- In the methods, it states AKI will be defined using the 2012 KDIGO criteria. Is this inclusive of both urine output and creatinine? If urine – how will this be quantified in incontinent children without indwelling bladder catheters.

- It states serum creatinine will be collected on admission and then at 48 hours post recruitment, but diagnosis of AKI is up to 72 hours. A creatinine will then have to be collected at 72 hours after admission. Please clarify this

- A modified renal angina index is mentioned in the background, but in the methods, this is not described. Please clarify.

Introduction:

- On page 5, line 98, it states that the development of CKD and ESKD are life limiting because of the constraints in access to KRT. Is it possible that these constraints also exist because of the resources available for transplant? Please add something about this.

- On line 98-100, it states genetic and immune risk factors – this is very specific and should be also stated this way in the abstract, and perhaps even modify the title accordingly.

- Line 103 – what is meant by “asexual forms”

- Line 106 – this belongs in the methods.

- Can the authors state why they think the risk factors for malaria associated AKI in Zambian children might be different from other countries in Africa?

- There are scattered sentences throughout the introduction of what the investigators intend to do. It would be helpful if this could be summated in a single few sentences.

- The overall introduction seems a bit disjointed with the introduction of a new topic with each paragraph. Perhaps a better approach would be to discuss the clinical charactertisics and how the renal angina index could be modified for better AKI prediction, and then enhance that prediction by also integrating patient specific genotypes and immune testing, as well as the biomarkers to refine the prediction of AKI.

Methods:

- In the study objectives, it states that they will use the renal angina index for AKI prediction. While it may be that the authors will show that its performance is not as good as in other critically ill children, I think they should also state that its modification with population specific predictors will lead to enhanced performance of AKI prediction.

- Please define what is meant by a level II mixed intensive care unit.

- The authors should consider how the onset of AKI might have occurred for those who are referred to UTH-CH, and the implications of their testing.

- It would be helpful to define a list of inclusion and exclusion up front. It was not realized until later that children less than 6 months will be excluded.

- On line 244, please specify whether these KDIGO criteria will include UOP – and referring to my comment in the intro, how this is handled in incontinent children with urine/stool mixes.

- The inclusion/exclusion criteria should be moved up above the definitions.

- There is a mix of tense in the methods section – some of this is described in past tense, and some in future tense, and this should be consistent.

- There are different sample sizes for each of the sub-analyses. Please provide a summative planned recruitment number with sufficient power. How will the renal angina index sample be determined?

- Again in the methods there is inconsistency with earlier statements on when AKI will be defined – 48 or 72 hours.

- On line 297 and 298, this is the only area where it seems that only serum creatinine will be used for the definition.

- Will laboratory tests be standardized across all patients?

- The renal outcome as the secondary outcome needs a little more clarity in its definition on line 343 on page 17. This is a bit confusing as written.

- The description of how the RAI will be modified to best suit the population is not described.

- NGAL measurement is in the blood? Since it is found in neutrophils, would expect this to be elevated in many patients, not just with AKI. Have the authors considered uNGAL. Same is true for KIM-1, both of which indicate tubular dysfunction and will be present in the urine in patients with AKI.

- The way in which each of the sections of the methods are defined should follow the same order as earlier in the methods.

- It is not clear why there are different patients being recruited for different portions of the study. If this is not true, there needs to be increased clarity on this.

- All the figures are blurry and cannot be commented on.

Reviewer #2: Chisambo et al shares a study protocol for an ongoing study to understand the burden, and risk factors for malaria associated acute kidney injury (MAKI)in children in Zambia. AKI is of growing importance in children in malaria endemic countries, and is a risk factor mortality Research to understand the pathogenesis are urgently needed.

Throughout the write up, there are several grammatical errors and loose/careless use of terms that the authors should correct

Abstract

A

Consider changing “escalates the risk of mortality sixfold”to increases the risk

and

93 “its long-term repercussions include impaired cognitive function and chronic kidney disease

94 (CKD), a phenomenon previously believed to be rare “. It is not clear which phenomenon the author is referring to here- CKD? OR AKI – both have previously been believed to be rare in children.

Methods

For objective 1and 2, clarify if the study population is <16-year s or 6months to 16-years. Reason for cut off at 16 years not clear?

Malaria pathogenesis and risk of complications including AKI, haemoglobinuria, severe anaemia etc. varies with age. There is a recent rise in the prevalence of AKI among older children, and indeed of hemoglobinuria in children older than 5 years, and age is am important factor here that authors need to consider-- what is if the majority of the cases are below five while controls are above 5?

The main issue is the study design for aims 1 and 2. An unmatched case control study is not the appropriate design to delineate risk factors for MAKI. Reasons for not match by age are not clear – Enroll children ages 6 to 16 years – match them by age: cases: AKI and controls non-AKI – Or at least by age groups. Given that at least 20-50% of children with severe malaria will develop AKI, these numbers can easily be estimated and matched

Additionally, all children referred to the study site with a diagnosis of MAKI already made will also be recruited as cases. Please clarify why you should depend on a referral diagnosis for enrollment

Study variables:

Renal outcome as a secondary outcome – This is presented here for the first time, out of nowhere, and needs a clearer definition.

Detailed description of hydration status, fluid balances with inputs and out puts, are important for MAKI .Further, suggest description of medicines used - e.g renal toxic medications to be collected between the groups as these are a potential but common risk factor for MAKI. How will severity fo MAKI be categorized ?

Further laboratory results on admission include HRPC

365 levels, - correct HRPC

What does biomolecule mean as mentioned in several sentences in the protocol?

Use of NGAL for as a biomarker of kidney injury is commendable

The Control group will be further categorized into subgroups, including cerebral malaria,

390 severe anemia, and uncomplicated malaria-? Not sure the study will be powered to see any differences. Further, what is the rationale? Why will children with un complicated malaria be followed for 3 days? What is the rationale for sun cataregorising into CM, SMA but not other severe malaria groups? How will these groups be defined

7. PLOS authors have the option to publish the peer review history of their article (what does this mean?). If published, this will include your full peer review and any attached files.

Reviewer #1: No

Reviewer #2: No

---

## [Author Response · Author response to Decision Letter 0]

10 Jul 2024

Dear Dr Conroy, 

Thank you for considering our manuscript and for the feedback received thus far.

Please find below a point-by-point response to the editor’s and reviewers’ comments.

Editor’s comments:

1 Editor comment: At present, the manuscript is organized as a proposal to an IRB or higher education committee and does not align with standard approaches to write up study protocols. For example, rather than presenting the null hypotheses, objectives, and sample size calculations as bullet points and separate sections, I encourage the authors to summarize the information and present it instead as hypotheses, and objectives, and present the sample size calculation that was used to decide the final sample size. 

Response: Done

2 Editor comment: Major revisions in the structuring of the content will be required for further consideration with the additional details requested by the reviewers. Please review the criteria on the website for guidance on how to structure protocol submissions and review published protocols for examples on what information to include. These guidelines must be followed for further consideration.

Response: Done

3 Editor comment: While an observational study does not have specific checklists for submission like clinical trials or systematic reviews, I strongly encourage the authors to review the SPIRIT and PRISMA checklists for study protocols and the STROBE guidelines on presenting results for observations studies to understand requirements for presentation of information

Response: Done. 

The STROBE checklist for case-control studies submitted alongside the manuscript

4 Editor comment: Please ensure that your manuscript meets PLOS ONE's style requirements, including those for file naming.

Response: Done

5 Editor comment: Your ethics statement should only appear in the Methods section of your manuscript. If your ethics statement is written in any section besides the Methods, please delete it from any other section.

Response: Done

Reviewer #1,s

1 Reviewer’s comment: Title:- To have “risk factors” and “predictors” in the title seems redundant, would select one of them.

Response: Title changed to:

“Clinical, immune and genetic risk factors of malaria-associated acute kidney injury in Zambian Children: A study protocol”

2 Reviewer’s comment: Abstract: - In the methods, it states AKI will be defined using the 2012 KDIGO criteria. Is this inclusive of both urine output and creatinine? If urine – how will this be quantified in incontinent children without indwelling bladder catheters.

Response: The urine output criterion of KDIGO 2012 will not be used in this study because the children that constitute the study population are not routinely monitored for urine output unless they are admitted to PICU.

Protocol amended to show this.

3 Reviewer’s Comment: Abstract: It states serum creatinine will be collected on admission and then at 48 hours post recruitment, but diagnosis of AKI is up to 72 hours. Creatinine will then have to be collected at 72 hours after admission. Please clarify this

Response: creatinine collection occurs at recruitment within the first 24 hours after admission (day 1) and then again on Day 3 and then finally at either discharge or day 7 whichever comes sooner.

Protocol amended with clearer phrasing

4 Reviewer’s comment: Introduction: A modified renal angina index is mentioned in the background, but in the methods, this is not described. Please clarify.

Response: This is presented under data analysis for sub-study 2:

“Local risk factors as identified from the cohort using multivariate logistic regression will be incorporated into the RAI in order to test if this will improve performance. Youden’s index will be used to determine the cut-off point for the modified RAI and curve fitting techniques will be utilized as described in previous studies by Matsuura et al. (57, 58,59) The AUC of the modified RAI and the AUC obtained from the original RAI will be compared using the z-test (non-parametric).”

References:

57. Matsuura R, Srisawat N, Claure-Del Granado R, Doi K, Yoshida T, Nangaku M, et al. Use of the Renal Angina Index in Determining Acute Kidney Injury. Kidney Int Rep. 2018;3(3):677-83.

58. Matsuura R, Iwagami M, Moriya H, Ohtake T, Hamasaki Y, Nangaku M, et al. A Simple Scoring Method for Predicting the Low Risk of Persistent Acute Kidney Injury in Critically Ill Adult Patients. Sci Rep. 2020;10(1):5726.

59. Zulu C, Mwaba C, Wa Somwe S. The renal angina index accurately predicts low risk of developing severe acute kidney injury among children admitted to a low-resource pediatric intensive care unit. Ren Fail. 2023;45(2):2252095. doi: 10.1080/0886022X.2023.2252095. 

5 Reviewer’s comment: Introduction: - On page 5, line 98, it states that the development of CKD and ESKD are life limiting because of the constraints in access to KRT. Is it possible that these constraints also exist because of the resources available for transplant? Please add something about this.

Response: sentence re-written:

The development of CKD and end-stage kidney disease (ESKD) in these patients, is life-limiting due to constraints in access to kidney replacement therapy and transplantation similar to other low-resource environments. (10)

6 Reviewer’s comment: Introduction: On line 98-100, it states genetic and immune risk factors – this is very specific and should be also stated this way in the abstract, and perhaps even modify the title accordingly.

Response: Title changed to:

“Clinical, immune and genetic risk factors of malaria-associated acute kidney injury in Zambian Children: A study protocol”

7 Reviewer’s comment: Introduction: Line 103 – what is meant by “asexual forms”

Response: sentence modified to read “asexual parasite forms(merozoites, schizonts)”

Sentence deleted based on reviewer comment number 8.

8 Reviewer’s comment: Introduction: Line 106 – this belongs in the methods.

Response: the sentence in line 106 i.e

“For this study MAKI will be defined as evidence of the presence of malaria asexual forms in a child meeting the KIDGO 2012 acute kidney injury (AKI) criteria.”

Has been deleted from introduction.

9 Reviewer’s comment: Introduction: Can the authors state why they think the risk factors for malaria associated AKI in Zambian children might be different from other countries in Africa?

Response: We plan to examine clinical, immune and genetic risk factor of MAKI in Zambian children.

Social-economic conditions in a given geographical location may alter the importance of previously described clinical risk factors. E.g local community propensity to use traditional healers prior to presentation may affect importance of herbal medications as a risk for MAKI, speed of seeking conventional health care.

Local medical practices may differ, and this may affect MAKI risk e.g. use of intravenous fluids, access to gentamicin. Access to organ support may further alter patient mortality rates.

Also, malaria transmission intensity alters population level immunity to malaria and this in turn may alter age groups affected by particular malaria syndromes. 

Genetic predisposition e.g. The importance of various single nucleotide polymorphisms (SNPs) in risk to severe malaria varies between various ethnic groups. 

Genetic makeup and distribution of SNPs across sub-Saharan Africa is diverse and previous studies have demonstrated that SNPs found to confer either susceptibility or protection to malaria / severe malaria syndromes in one part of Africa are not necessarily important when tested in other populations on the continent.

Similarly, because the function of immune proteins such as cytokines is determined by genes, the aforementioned genetic diversity in SNPs may result in varying importance of these immune proteins in the pathogenesis of MAKI in different ethnic groups.

The points cited above suggest that we need multicenter studies that include multiple ethnicities and geographical areas to further establish/confirm potential risk factors in MAKI. 

Manuscript amended to include this sentence:

Page 5Line 101: “The importance of any particular risk factor in any given population may be modulated by peculiarities in local medical practice, access to healthcare resources, social determinants of health, changing malaria transmission intensity patterns and the effect of genetic variability on the function of the various immune molecules that are involved in MAKI pathogenesis.”

10 Reviewer Comment: Introduction: - There are scattered sentences throughout the introduction of what the investigators intend to do. It would be helpful if this could be summated in a single few sentences.

Response: Done

11 Reviewer’s comment: Introduction: The overall introduction seems a bit disjointed with the introduction of a new topic with each paragraph. Perhaps a better approach would be to discuss the clinical charactertisics and how the renal angina index could be modified for better AKI prediction, and then enhance that prediction by also integrating patient specific genotypes and immune testing, as well as the biomarkers to refine the prediction of AKI.

Response: Introduction revised to try and improve the flow as suggested.

12 Reviewer’s comment: Methods: - In the study objectives, it states that they will use the renal angina index for AKI prediction. While it may be that the authors will show that its performance is not as good as in other critically ill children, I think they should also state that its modification with population specific predictors will lead to enhanced performance of AKI prediction.

Response: Done. Specific objective 4 modified to:

“To determine the utility of the Renal Angina Score in predicting malaria-associated AKI and access whether its modification with population specific predictors will lead to enhanced performance of AKI prediction”

13 Reviewer’s comment: methods: Please define what is meant by a level II mixed intensive care unit.

Response: Sentence re-written:

“The paediatrics department has a capacity of 70 beds with access to level II intensive care unit (ICU) that admits neonates, children and adults with both medical and surgical conditions.”

14 Reviewer’s comment: Methods: The authors should consider how the onset of AKI might have occurred for those who are referred to UTH-CH, and the implications of their testing.

Response: Noted. We are collecting information on clinical characteristics like presence of diarrhea, vomiting, use of herbal medications, dose and type of drugs administered at home or the referral center. If blood results from the referring clinic are available these will be examined as well

15 Reviewer’s comment: Methods: - It would be helpful to define a list of inclusion and exclusion up front. It was not realized until later that children less than 6 months will be excluded.

Response: Done

16 Reviewer’s comment: Methods: - On line 244, please specify whether these KDIGO criteria will include UOP – and referring to my comment in the intro, how this is handled in incontinent children with urine/stool mixes.

Response: The study population includes children whose urine output is not routinely monitored and unless they are admitted to PICU are not catheterized.

For this reason, the UOP KDIGO 2012 criterion will not be used

17 Reviewer’s comment: Methods: The inclusion/exclusion criteria should be moved up above the definitions.

Response: Done

18 Reviewer’s comment: Methods: There is a mix of tense in the methods section – some of this is described in past tense, and some in future tense, and this should be consistent.

Response: Done

19 Reviewer’s comment: Methods: There are different sample sizes for each of the sub-analyses. Please provide a summative planned recruitment number with sufficient power. How will the renal angina index sample be determined?

Response: Done

For recruitment into the RAI study: Any participant with normal creatinine at admission (does not meet KDIGO creatinine criteria), will be recruited into the sub-study 2 (RAI) study. Page 22 line 411-415

The details on sample size estimation have been placed in supplementary file 1:

Sample size for the RAI study was determined using the Cochrane formula:

Where n = the required sample size

Z = Normal Standard Deviation taken with a 95% Confidence Interval; set at 1.96.

d = margin of error (0.05)

p = prevalence of AKI in malaria 

Assumptions made were a significance level of 0.05, a prevalence of AKI in children admitted with malaria of 0.30 and power of 80%.

Allowing for a 10% loss of data a total sample size of 180 children with malaria and no AKI at admission to the hospital, will need to be recruited into the RAI study. 

s20 Reviewer’s comment: Methods: Again, in the methods there is inconsistency with earlier statements on when AKI will be defined – 48 or 72 hours.

Response: Participants are enrolled within the first 24 hours post admission (day 1) and blood is tested again for creatinine 48 hours later (Day 3).

Protocol harmonized.

21 Reviewer’s comment: Methods: On line 297 and 298, this is the only area where it seems that only serum creatinine will be used for the definition.

Response: The AKI criteria used has been clarified and harmonized throughout the manuscript.

Because this is a general hospital population urine output is not routinely monitored. Thus, for this study the urine output criterion of the KIDGO 2012 will not be used to define AKI.

Only the KIDGO 2012 creatinine criterion is used

22 Reviewer’s comment: Methods: Will laboratory tests be standardized across all patients?

Response: Yes. Specimens for chemistry, immunology will be frozen at -80 C while genetic samples will be collected on dried blood spots. The tests will be performed at UTH at either the KS molecular laboratory or UNZA veterinary school genomics laboratory or the UTH chemistry laboratory respectively.

23 Reviewer’s comment: Methods: The renal outcome as the secondary outcome needs a little more clarity in its definition on line 343 on page 17. This is a bit confusing as written.

Response: Amended to read

“Serum creatinine will be collected at discharge or if the patient is still hospitalized on day 7, whichever comes sooner, in order to establish the renal outcome at day 7. Renal outcome will be noted as a secondary outcome. Outcome will be classified as normal if eGFR is > 120ml/1.73m2/min or abnormal for eGFR < X. The Schwartz formula will be used to estimate GFR. “

24 Reviewer’s comment: Methods: The description of how the RAI will be modified to best suit the population is not described.

Response: The statement below is placed in the data analysis plan for sub-study 2

“Local risk factors as identified from the cohort using multivariate logistic regression will be incorporated into the RAI in order to test if this will improve performance. Youden’s index will be used to determine the cut-off point for the modified RAI and curve fitting techniques will be utilized as described in previous studies by Matsuura et al. (57, 58) The AUC of the modified RAI and the AUC obtained from the original RAI will be compared using the z-test (non-parametric).”

References:

57. Matsuura R, Srisawat N, Claure-Del Granado R, Doi K, Yoshida T, Nangaku M, et al. Use of the Renal Angina Index in Determining Acute Kidney Injury. Kidney Int Rep. 2018;3(3):677-83.

58. Matsuura R, Iwagami M, Moriya H, Ohtake T, Hamasaki Y, Nangaku M, et al. A Simple Scoring Method for Predicting the Low Risk of Persistent Acute Kidney Injury in Critically Ill Adult Patients. Sci Rep. 2020;10(1):5726.

59. Zulu C, Mwaba C, Wa Somwe S. The renal angina index accurately predicts low risk of developing severe acute kidney injury among children admitted to a low-resource pediatric intensive care unit. Ren Fail. 2023;45(2):2252095. doi: 10.1080/0886022X.2023.2252095. 

25 Reviewer’s comment: Methods: NGAL measurement is in the blood? Since it is found in neutrophils, would expect this to be elevated in many patients, not just with AKI. Have the au

---

## [Decision Letter · Decision Letter 1]

3 Sep 2024

PONE-D-24-13075R1Clinical, immune and genetic risk factors of malaria-associated acute kidney injury in Zambian Children: A study protocolPLOS ONE

Dear Dr. Mwaba,

Thank you for submitting your manuscript to PLOS ONE. After careful consideration, we feel that it has merit but does not fully meet PLOS ONE’s publication criteria as it currently stands. Therefore, we invite you to submit a revised version of the manuscript that addresses the points raised during the review process.

We look forward to receiving your revised manuscript.

Kind regards,

Andrea L. Conroy, PhD

Academic Editor

PLOS ONE

Additional Editor Comments:

Thank your for your attention to the reviewers comments and work in revising the protocol. Both reviewers recognized the improvements in the manuscript but felt like there were still issues with the writing that needed to be addressed and some bigger picture inconsistencies regarding laboratory measures between sections of the protocol. As this study is currently recruiting, I strongly recommend that you use this feedback and critically review the protocol in detail to ensure each section is internally consistent and matches what is happening on the ground with recruitment. This will be an important resource for others looking to validate your work, and will help with the ultimate write up and publication of the study findings.

I have reviewed the comments from the reviewers and consider all of their comments appropriate. Please address them point-by-point and provide an overview of what changes were made to ensure each section of the protocol is consistent.

Reviewers' comments:

Reviewer's Responses to Questions

**Comments to the Author**

1. Does the manuscript provide a valid rationale for the proposed study, with clearly identified and justified research questions?

Reviewer #1: Partly

Reviewer #2: Yes

2. Is the protocol technically sound and planned in a manner that will lead to a meaningful outcome and allow testing the stated hypotheses?

Reviewer #1: Partly

Reviewer #2: Yes

3. Is the methodology feasible and described in sufficient detail to allow the work to be replicable?

Reviewer #1: Yes

Reviewer #2: Yes

4. Have the authors described where all data underlying the findings will be made available when the study is complete?

Reviewer #1: Yes

Reviewer #2: Yes

5. Is the manuscript presented in an intelligible fashion and written in standard English?

Reviewer #1: No

Reviewer #2: No

6. Review Comments to the Author

You may also provide optional suggestions and comments to authors that they might find helpful in planning their study.

Reviewer #1: Risk factors and predictors of malaria-associated acute kidney injury in Zambian

Children: A study protocol

The authors describe the methods for a prospective observational study in which they will match malaria associated AKI patients to those without malaria associated AKI, occurring by day 3 from admission.

The protocol in its revised version is improved, although there are still many areas that area confusing and require revision for improved clarity. There continues to be inconsistency in when some laboratory measures will be obtained, varying between the abstract and body of the protocol.

Title:

- Much improved, no further ocmments

Abstract:

- In the responses, the authors state creatinine will also be measured on day 7 or discharge, whichever is sooner, but this is not mentioned in the section on collection of other creatinine tests.

- The utilization of the renal angina index should precede the brief description of statistical analysis.

-

Introduction:

- This is substantially more succinct and clear that this will be a study protocol, and provides a construct for what is to come.

- Line 99 – this is a very long sentence that’s starts and ends with MAKI. Please revise for improved grammar separating risk factors and pathogenesis.

- Line 131 – the word cytoadherence is present 3 times in tis sentence, please revise for improved grammar.

- In the responses, there are multiple references provided for the renal angina index. This modification for malaria and in low resource settings should be included and cited in the introduction.

- Line 158 (revised version) AKI and MAKI are in the same sentence, please revise for improved grammar.

- Hypothesis 3 states renal angina – is this the original one, or a modified version, please be specific and reference which is to be used.

- In aim 3, it states the biomarkers will be measured at 48 hours – but earlier, and in the figure it states day 3. This should be consistent.

Methods:

- Line 205 of when malaria occurs, perhaps this fits better in the introduction.

- The sentence on line 209 “the paeditric department has a capacity…” is repeated, please correct.

- The methods section is still a bit confusing to follow and could be improved.

- Line 355 – the Schwartz formula should be referenced. Is this the standard one with varying factors, or the modified bedside schwartz?

-

-

Reviewer #2: The authors have made great progress in revising the manuscript, and have responded to most of the reviewer comments. There are scattered grammatical errors throughout the manuscript, including spacing, misplaced comas and spellings that need to be corrected before publication. The authors should take a keen look at the manuscript and correct all the errors.

7. PLOS authors have the option to publish the peer review history of their article (what does this mean?). If published, this will include your full peer review and any attached files.

Reviewer #1: No

Reviewer #2: No

---

## [Author Response · Author response to Decision Letter 1]

4 Oct 2024

Dear Dr Conroy, 

Thank you for the feedback.

Please find below a point-by-point response to the editor’s and reviewers’ comments.

Editor’s:

1 Editor comment: Thank you for your attention to the reviewers’ comments and work in revising the protocol. Both reviewers recognized the improvements in the manuscript but felt like there were still issues with the writing that needed to be addressed and some bigger picture inconsistencies regarding laboratory measures between sections of the protocol. As this study is currently recruiting, I strongly recommend that you use this feedback and critically review the protocol in detail to ensure each section is internally consistent and matches what is happening on the ground with recruitment. This will be an important resource for others looking to validate your work and will help with the ultimate write up and publication of the study findings.

Response: Done

2 Editor comment: I have reviewed the comments from the reviewers and consider all of their comments appropriate. Please address them point-by-point.

Response: Done

3. Editor comment: Provide an overview of what changes were made to ensure each section of the protocol is consistent.

Response: The language between the abstract and the methods section were harmonized. Throughout we now use “day” instead of hours to describe points at which serum creatinine will be collected in order to be consistent. Figure 1 further clarifies recruitment procedures.

The abstract has been revised to show that Day 7/discharge bloods are also being collected and that the study includes a nested prospective observational study.

Additionally, we added guidepost sentences such as:

- on page 12, line 256:

“In the following sections the recruitment and laboratory procedures, the study variables, and the data analysis plans for sub-study 1 (section 2.6) and the nested prospective sub-study 2 (section 2.7) have outlined.”

-page 8, line 179:

“The project is divided into two sub-studies, namely 1) Sub-study 1 which answers objectives 1, 2 and 3, and is an unmatched case-control study with a case:control ratio of 1:1. An unmatched design was chosen because this is an exploratory study. However, during analysis of data for the immune molecules, patients will be analysed based on age groups. Sub-study 2 answers objectives 4 and 5 and is a nested prospective observational study that recruits participants from the control arm of sub-study 1 (Fig1).“

Reviewer #1,s

1 Reviewer’s comment: Overall: The protocol in its revised version is improved, although there are still many areas that are confusing and require revision for improved clarity. There continues to be inconsistency in when some laboratory measures will be obtained, varying between the abstract and body of the protocol.

Response: protocol harmonized

2 Reviewer’s comment: Title: Much improved, no further comments

Response: Thank you

3 Reviewer’s Comment: Abstract: In the responses, the authors state creatinine will also be measured on day 7 or discharge, whichever is sooner, but this is not mentioned in the section on collection of other creatinine tests.

Response: The statement below added to abstract: Page 4, line 60-64:

“AKI is defined using the 2012 KIDGO AKI creatinine criteria, and cases are defined as children admitted with malaria who develop AKI within 72 hours of admission, while controls are children admitted with malaria but with no AKI. Serum creatinine is collected on Day 1 within 24 hours of admission, on Day 3 and then again on discharge or day 7, whichever comes sooner. ”

4 Reviewer’s comment: Abstract: The utilization of the renal angina index should precede the brief description of statistical analysis

Response: the statement has been modified to read:

 “The sensitivity, specificity, and estimates of the area under the curve (AUC) for the renal angina score will be determined.”

5 Reviewer’s comment: Introduction: This is substantially more succinct and clear that this will be a study protocol, and provides a construct for what is to come.

Response: Thank you

6 Reviewer’s comment: Introduction: Line 99 – this is a very long sentence that’s starts and ends with MAKI. Please revise for improved grammar separating risk factors and pathogenesis.

Response: page 5: line 92-94:

‘Risk factors in MAKI can be categorized as either biological or clinical. Biological risk factors are the various molecules involved in MAKI pathogenetic mechanisms such as cytoadherence, immune dysregulation, heme nephrotoxicity, and increased oxidative stress.[6, 7, 11, 12, 13] “

7 Reviewer’s comment: Introduction: - Line 131 – the word cytoadherence is present 3 times in this sentence, please revise for improved grammar.

Response: line 124-127: revised to read:

Furthermore, cytoadherence causes endothelial activation, which in turn causes cytoadherence to worsen. Thus, a vicious cycle is established in which endothelial activation results from cytoadherence and vice versa. [11, 13, 25]

8 Reviewer’s comment: Introduction: In the responses, there are multiple references provided for the renal angina index. This modification for malaria and in low resource settings should be included and cited in the introduction.

Response: This sentence added to introduction: page 7: line 144-145

“Some studies have attempted to modify the RAI for use in different populations. [40,41,42]” 

9 Reviewer’s comment: Introduction: - Line 158 (revised version) AKI and MAKI are in the same sentence, please revise for improved grammar.

Response: Sentence modified to read:

“We hypothesize that various clinical factors, serum concentration of immunological molecules, as well as genetic polymorphisms of selected immune factors are correlated to the risk of developing AKI in children admitted with malaria and can therefore be used for risk prediction.”

10 Reviewer Comment: Introduction: Hypothesis 3 states renal angina – is this the original one, or a modified version, please be specific and reference which is to be used.

Response: At this point we have not yet derived the parameters for the modified RAI as that will only be done once the data has been collected and regression analysis done.

Sentence modified to read: page 8: line 165:

“3) that the Renal Angina Index and/or its modification, NGAL, suPAR and KIM-1 when used in severe malaria may be useful in predicting severe AKI two days later (Day 3)”

11 Reviewer’s comment: Introduction: In aim 3, it states the biomarkers will be measured at 48 hours – but earlier, and in the figure it states day 3. This should be consistent.

Response: modified to:

“3) that the Renal Angina Index and/or its modification, NGAL, suPAR and KIM-1 when used in severe malaria may be useful in predicting severe AKI two days later (Day 3)”

12 Reviewer’s comment: Methods: Line 205 of when malaria occurs, perhaps this fits better in the introduction.

Response: sentence moved to section on study limitations

13 Reviewer’s comment: Methods: The sentence on line 209 “the paediatric department has a capacity…” is repeated, please correct.

Response: Done

14 Reviewer’s comment: Methods: - The methods section is still a bit confusing to follow and could be improved.

Response: Done. Added guidepost sentences such as:

- on page 12, line 256:

“In the following sections the recruitment and laboratory procedures, the study variables, and the data analysis plans for sub-study 1 (section 2.6) and the nested prospective sub-study 2 (section 2.7) are outlined.”

-page 8, line 179:

“The project is divided into two sub-studies, namely 1) Sub-study 1 which answers objectives 1, 2 and 3, and is an unmatched case-control study with a case:control ratio of 1:1. An unmatched design was chosen because this is an exploratory study. However, during analysis of data for the immune molecules, patients will be analysed based on age groups. Sub-study 2 answers objectives 4 and 5 and is a nested prospective observational study that recruits participants from the control arm of sub-study 1 (Fig1).“

15 Reviewer’s comment: Methods: Line 355 – the Schwartz formula should be referenced. Is this the standard one with varying factors, or the modified bedside schwartz?

Response: Bedside. Reference provided:

56. Schwartz GJ, Haycock GB, Edelmann CM, Jr., Spitzer A. A simple estimate of glomerular filtration rate in children derived from body length and plasma creatinine. Pediatrics. 1976;58(2):259-63.

Reviewer #2

1 Reviewer’s comment: The authors have made great progress in revising the manuscript and have responded to most of the reviewer comments. There are scattered grammatical errors throughout the manuscript, including spacing, misplaced comas and spellings that need to be corrected before publication. The authors should take a keen look at the manuscript and correct all the errors.

Response: Grammatical errors corrected.

Sincerely, 

Chisambo Mwaba, BSc (Human Biology), MBChB, Mmed, Cert. Paed. Neph. (SA), MPhil 

Lecturer 

University of Zambia Medical School

---

## [Decision Letter · Decision Letter 2]

8 Dec 2024

Clinical, immune and genetic risk factors of malaria-associated acute kidney injury in Zambian Children: A study protocol

PONE-D-24-13075R2

Dear Dr. Mwaba,

We’re pleased to inform you that your manuscript has been judged scientifically suitable for publication and will be formally accepted for publication once it meets all outstanding technical requirements.

Kind regards,

Steve Zimmerman, PhD

Senior Editor, PLOS ONE

Additional Editor Comments (optional):

Reviewers' comments:

Reviewer's Responses to Questions

**Comments to the Author**

1. Does the manuscript provide a valid rationale for the proposed study, with clearly identified and justified research questions?

Reviewer #1: Yes

Reviewer #2: Yes

2. Is the protocol technically sound and planned in a manner that will lead to a meaningful outcome and allow testing the stated hypotheses?

Reviewer #1: Yes

Reviewer #2: Yes

3. Is the methodology feasible and described in sufficient detail to allow the work to be replicable?

Reviewer #1: Yes

Reviewer #2: Yes

4. Have the authors described where all data underlying the findings will be made available when the study is complete?

Reviewer #1: Yes

Reviewer #2: Yes

5. Is the manuscript presented in an intelligible fashion and written in standard English?

Reviewer #1: Yes

Reviewer #2: Yes

6. Review Comments to the Author

You may also provide optional suggestions and comments to authors that they might find helpful in planning their study.

Reviewer #1: The authors have addressed all my comments. The manuscript is substantially improved, having now reviewed 3 versions. I very much look forward to reading the results of the study and how this can chang and improve care in the future.

Reviewer #2: I have no additional comments for the authors . The authors have done a good job at revising the manuscript

7. PLOS authors have the option to publish the peer review history of their article (what does this mean?). If published, this will include your full peer review and any attached files.

Reviewer #1: **Yes: **Katja M Gist

Reviewer #2: No

---

## [Editor Report · Acceptance letter]

12 Dec 2024

PONE-D-24-13075R2 

PLOS ONE

Dear Dr. Mwaba, 

I'm pleased to inform you that your manuscript has been deemed suitable for publication in PLOS ONE. Congratulations! Your manuscript is now being handed over to our production team.

Kind regards, 

on behalf of

Dr Steve Zimmerman 

Staff Editor

PLOS ONE